# Genome-Wide Identification and Expression Analysis of Homeodomain Leucine Zipper Subfamily IV (HD-ZIP IV) Gene Family in *Cannabis sativa* L.

**DOI:** 10.3390/plants11101307

**Published:** 2022-05-13

**Authors:** Gang Ma, Alice Kira Zelman, Peter V. Apicella, Gerald Berkowitz

**Affiliations:** 1Agricultural Biotechnology Laboratory, Department of Plant Science and Landscape Architecture, University of Connecticut, Storrs, CT 06269, USA; gang.ma@uconn.edu (G.M.); ceilamanzelcanth@gmail.com (A.K.Z.); papicella@mydecineinc.com (P.V.A.); 2Mydecine Innovations Group Inc., Denver, CO 80231, USA

**Keywords:** *Cannabis sativa* L., homeodomain zipper IV transcription factors, bioinformatics, gene regulation, expression pattern, flower maturity, trichomes

## Abstract

The plant-specific homeodomain zipper family (HD-ZIP) of transcription factors plays central roles in regulating plant development and environmental resistance. HD-ZIP transcription factors IV (HDZ IV) have been involved primarily in the regulation of epidermal structure development, such as stomata and trichomes. In our study, we identified nine HDZ IV-encoding genes in *Cannabis sativa* L. by conducting a computational analysis of cannabis genome resources. Our analysis suggests that these genes putatively encode proteins that have all the conserved domains of HDZ IV transcription factors. The phylogenetic analysis of HDZ IV gene family members of cannabis, rice (*Oryza sativa*), and *Arabidopsis* further implies that they might have followed distinct evolutionary paths after divergence from a common ancestor. All the identified cannabis HDZ IV gene promoter sequences have multiple regulation motifs, such as light- and hormone-responsive elements. Furthermore, experimental evidence shows that different HDZ IV genes have different expression patterns in root, stem, leaf, and flower tissues. Four genes were primarily expressed in flowers, and the expression of *CsHDG5* (XP_030501222.1) was also correlated with flower maturity. Fifty-nine genes were predicted as targets of HDZ IV transcription factors. Some of these genes play central roles in pathogen response, flower development, and brassinosteroid signaling. A subcellular localization assay indicated that one gene of this family is localized in the *Arabidopsis* protoplast nucleus. Taken together, our work lays fundamental groundwork to illuminate the function of cannabis HDZ IV genes and their possible future uses in increasing cannabis trichome morphogenesis and secondary metabolite production.

## 1. Introduction

As seeds grow into mature plants, precise gene transcription and translation regulation are crucial in every biological process, including growth, development, and metabolism [1]. Transcription factors play crucial roles in the transcriptional regulation of gene expression [2,3]. They recognize and bind to specific regulatory elements in the promoter regions of target genes [4]. The expression of these target genes could be modulated at the transcriptional level based on the need for growth or resistance to environmental stresses [5]. Thus, the identification and functional analysis of transcription factor genes are desired for understanding fundamental knowledge about the molecular mechanisms of various biological processes [6]. Transcription factors have different binding domains, which could diversify into multiple families [7,8]. Transcription factors with homeodomain-zipper (HDZ) domains are specific to plant species [9,10]. The HDZ gene family has been classified into four subfamilies from I to IV based on the presence and absence of additional conserved domains [11]. Specific domains only occur in specific HDZ subfamilies; for example, the MEKHLA domain only appears in the members of the HDZ III subfamily [12].

As whole genome sequencing technology develops, increasing numbers of plant species are being sequenced [13,14]. An analysis of the HDZ IV gene family has been carried out in different plants including *Arabidopsis*, rice, maize, soybean, and cucumber [15,16,17,18,19]. These genes are mainly expressed in epidermal cells of plant organs and in the outermost layer of the shoot apical meristem [15,16,17,18,19]. Recent works used RNA sequencing to show the functional characterization and expression patterns of HDZ IV transcription factors and how they regulate the initiation of epidermal structures, including stomata, trichomes, and root hairs as well as cuticle development [10]. Abundant mutant resources enabled the utilization of a deep functional analysis of members of the *Arabidopsis* HDZ IV gene family. *Arabidopsis* HDZ IV transcription factor GLABRA2 (*GL2*) plays important roles in trichome development and root hair patterning [20,21,22]. The roles of *Arabidopsis* PROTODERMAL FACTOR 2 (*AtPDF2*) and MERISTEM LAYER 1 (*AtML1*), two functionally redundant HDZ IV genes, have been clarified in terms of the regulation of epidermis and flower development [23]. ANTHOCYANINLESS 2 (*AtANL2*) is involved in sub-epidermal cells’ anthocyanin deposition, epidermal cell proliferation, and root growth [24,25,26]. One of the *Arabidopsis* HDZ IV transcription factors, AtHDG11, also plays a starring role in plant resistance to drought stress by improving root development and reducing stomatal density [27,28,29]. Functional analysis of some of the HDZ IV genes has also been carried out in other plant species such as *Artemisia annua* L. (sweet wormwood) and *Solanum lycopersicum* (tomato). In sweet wormwood, homeodomain protein 1 (*AaHD1*) regulates glandular trichome initiation. Interestingly, this gene is involved in the phytohormone jasmonate (JA)-mediated glandular trichome initiation [30]. Tomato homeodomain protein *Sl**PDF2* induces trichome formation and embryo lethality. The overexpression of *PDF2* in tomato leads to the generation of more trichomes in the shoot, leaf, and floral tissues [31]. From all these studies, it can be concluded that HDZ IV transcription factors are vital to plant-specific epidermal structures’ development. The investigation and functional analysis of this gene family in other crop plants is urgently needed to solve important application problems.

*Cannabis sativa* L., one of the first domesticated plants, is separated into fiber-based hemp and marijuana, from which both medicinal and recreational drugs are obtained [32]. Cannabis generates abundant secondary metabolites, for which their therapeutic potential, particularly in the cases of cannabidiolic acid (CBDA) and Δ9-tetrahydrocannabinolic acid, has renewed global attention [33]. One type of epidermal structure, glandular trichomes, produces known and unknown secondary metabolites [34]. To increase the content of these secondary metabolites, farmers and plant researchers have tried to optimize growth conditions by, for example, increasing light or improving nutrition [35]. Using various additives including phytohormones can increase secondary metabolites content in some specific conditions [36]. However, these methods cannot fundamentally change the situation. How then can we increase the number of glandular trichomes, especially in leaves? Plant molecular biology and functional analysis of specific genes could provide an answer. In this study, we used bioinformatics strategies to carry out genome-wide identification and functional analysis of HDZ IV transcription factors in cannabis. Furthermore, we also experimentally studied the expression patterns of individual genes of this family during flower development. From these two studies, we identify the HDZ IV transcription factor gene family in cannabis, which could be used as a possible solution to increase the number of cannabis trichomes and improve secondary metabolite content in future work.

## 2. Results

### 2.1. Identification of HDZ Subfamily IV Genes in Cannabis

The Arabidopsis (16 genes) and Oryza sativa (11 genes) HDZ subfamily IV transcription factor genes were used as queries in BLASTp searches against a cannabis genome database [37]. Redundant sequences, sequences without conserved homeodomain, and sequences with a MEKHLA domain (HDZ sub family III specific domain) were removed (there were three proteins containing MEKHLA domain analyzed by Pfam database and Conserved Domain Database) (Appendix A). A total of nine genes encoding putative HDZ IV genes in cannabis were identified. These cannabis HDZ IV proteins contained characteristic domains, namely homeobox and Steroidogenic Acute Regulatory (StAR)-related lipid Transfer (START). Various features of putatively identified cannabis HDZ IV genes, such as length, predicted protein molecular weight, isoelectric point (pI), grand average of hydropathicity (GRAVY), instability index, aliphatic index, and subcellular location, are summarized in Table 1. The molecular mass of cannabis HDZ IV proteins ranged from 79,965.59 to 91,400.22 kDa, with an average molecular mass of 85,846.23 kDa; the protein length ranged from 737 to 841 residues, with an average length of 781 aa; the isoelectric point of the HDZ IV proteins ranged from 5.62 to 6.41 with the average protein isoelectric point of 5.84. The grand average of hydropathicity (GRAVY) is a measure of its hydrophobicity or hydrophilicity. The GRAVY of cannabis HDZ IV proteins ranged from −0.458 to −0.282. The hydrophobic amino acids may affect the stabilization and function of cannabis HDZ IV proteins. The instability index of proteins represents whether the protein will be stable in a test tube. The instability index of cannabis HDZ IV proteins showed their range from 40.27 to 54.15, indicating that none of the cannabis HDZ IV proteins are stable in vitro. The aliphatic index of a protein is defined as the relative volume occupied by aliphatic side chains (alanine, valine, isoleucine, and leucine), which may be regarded as a positive indicator of thermostability of globular proteins. The predicted cannabis HDZ IV proteins’ aliphatic indices ranged from 74.75 to 87.14. Therefore, cannabis HDZ IV proteins may be stable under higher temperatures in vivo. The subcellular localization prediction showed cannabis HDZ IV proteins that were primarily located in the nucleus. Interestingly, two cannabis HDZ IV proteins, CsHDG5 (XP_030501222.1) and CsHDG5-like (XP_030501651.1), had nearly identical properties (Table 1) and sequences, and they may represent a recent gene duplication, although these sequences should be confirmed as distinct genes.

### 2.2. Phylogenetic Analysis of Cannabis HDZ IV Proteins

The phylogenetic tree was constructed using Maximum Likelihood method (execution parameter: bootstrap method 500) to understand the evolutionary relationships between cannabis HDZ IV proteins and model plant HDZ IV proteins (Arabidopsis and Oryza sativa) [38]. Most HDZ IV proteins from cannabis and the dicot model plant Arabidopsis clustered together (Figure 1).

### 2.3. Conserved Motifs in Cannabis HDZ IV Proteins

To understand the possible functions of cannabis HDZ IV proteins, a comprehensive prediction website/software was used to find conserved motifs [41]. A total of 10 conserved motifs were queried in nine cannabis HDZ IV proteins. Figure 2 and Appendix A show the details about conserved motifs in cannabis HDZ IV proteins. Motif 1 corresponded to the conserved Homeodomain (HD), and motifs 2, 6, 7, and 9 belonged to the START domain (Steroidogenic Acute Regulatory (StAR)-related lipid Transfer) [42].

### 2.4. Gene Structure and Chromosomal Localization of Cannabis HDZ IV Genes

Untranslated Regions (UTR) and introns play crucial roles in post-transcriptional regulation and alternative splicing, respectively [43]. It is important to gain information about cannabis HDZ IV gene structure, which could affect gene expression and translation. Figure 3 shows that the numbers of exons and introns among cannabis HDZ IV genes varied between 8 to 18 and 7 to 17, respectively. The lengths of UTR in cannabis HDZ IV genes were different, which means that their post-transcriptional regulation may also differ. Intriguingly, CsROC8-like (XM_030653335.1) did not have predicted 5′ and 3′ UTRs. The regulation pattern of this gene may be different from other HDZ IV genes.

Figure 4 exhibits the cannabis HDZ IV genes’ locations mapped to the cannabis chromosomes. No HDZ IV gene was present on chromosome 2, 3, 6, 7, or X. Chromosomes 1 and 5 contained three HDZ IV genes. There was one HDZ IV gene on chromosomes 4, 8, and 9, respectively.

### 2.5. Prediction and Analysis of Cannabis HDZ IV Genes Promoter

Promoter analysis is an effective strategy for understanding transcriptional regulation of specific genes. Two-thousand base pair upstream DNA sequences were collected using TBtools [44]. In Figure 5, we can see that the most frequently occurring regulatory domain is related to light. Light is the most important environmental factor in the regulation of the plant flowering stage [45,46]. Another interesting finding is that cannabis HDZ IV genes contain multiple hormone-responsive elements, suggesting that these genes can be regulated by multiple hormones [47].

### 2.6. Expression Pattern of Specific Cannabis HDZ IV Genes in Different Tissues and during Flower Maturation

To understand the expression pattern of this gene family, we isolated total RNAs from cannabis Space Candy (SC) variety root, stem, leaf, and flower tissues. Figure 6 shows that the expression of eight out of nine genes was detected in different tissues; the exception is ROC8-like (XP_030509195.1), which may be regulated by a specific stimulus [15]. Eight genes of the HDZ-IV family have different expression patterns in different tissues. All genes except PROTODERMAL FACTOR 2 (XP_030492336.1) were not expressed in the root. Therefore, we used expressions in the leaf as our control. There were four genes (HDG5-like, GLABRA 2, HDG5, and ANTHOCYANINLESS 2 isoform X2) primarily expressed in the flower tissues, which may be involved in trichome morphogenesis and the development in cannabis [19]. Interestingly, the expression of GLABRA 2 (XP_030491770.1) was 30-fold higher in the flower. The GLABRA 2 homolog in Arabidopsis was involved in trichome development and root hair patterning [20,21,22]. However, we could not detect GLABRA 2 (XP_030491770.1) expression in roots. Cannabis GLABRA 2 may have a specific function in trichome morphogenesis. HDG11 (XP_030499384.1), HDG2 (XP_030482406.1) and PROTODERMAL FACTOR 2 (XP_030492336.1) had low expression in these tissues.

*Artemisia annua* L. (sweet wormwood) has glandular trichomes that can generate the secondary metabolite artemisinin [48]. These glandular trichomes are similar to the glandular trichomes found in cannabis. The AaHD1 protein is important in glandular trichome initiation [30]. Cannabis homeobox-leucine zipper protein HDG5 (XP_030501222.1) is an ortholog of AaHD1. We wanted to know the function and expression pattern of CsHDG5. Figure 7 shows the expression of CsHDG5 during flower maturity (from week 1 to week 7). CsHDG5 was expressed at the highest level in week 3 and week 4; after that, its expression decreased. Furthermore, the functional analysis of CsHDG5 showed that CsHDG5 may initiate trichome development (unpublished data).

### 2.7. Putative Targets of HDZ IV Transcription Factors

It is important to know that HDZ IV transcription factors regulate substrates in cannabis. A previous study in Arabidopsis indicated that some HDZ IV transcription factors could bind to 5′-GCATTAAATGC-3′ consensus sequences [19]. We analyzed and identified 59 genes’ promoter sequences containing this motif. Figure 8 and Appendix A show that these 59 genes include 16 unknown/uncharacterized genes, 11 genes encoding enzymes, and 32 functional genes. XM_030645201.1 was predicted as an uncharacterized protein similar to At3g49140, which belongs to the pentatricopeptide repeat (PPR) superfamily of proteins [49]. The promoter sequence of XM_030645201.1 has three 5′-GCATTAAATGC-3′ consensus sequences. This gene may be one of the targets of HDZ IV transcription factors. Our unpublished data also indicate that one of the HDZ IV transcription factors was induced by Arabidopsis pathogen elicitor peptide 3 (AtPep3) in trichomes [50]. HDZ IV transcription factors may play crucial roles in plant responses to biotic stresses by regulating specific substrates [51]. We also found that XM_030630414.1, XM_030634601.1 and XM_030649628.1 were predicted to be involved in flower development [52,53]. Interestingly, a putative transcription factor, bHLH63, may be regulated by a specific HDZ IV transcription factor. Transcription factor bHLH63/CIB1 in Arabidopsis could bind a G-box or E-box to promote FT gene expression and, thus, trigger flowering in response to blue light [54].

Flower yield is vital in the cannabis industry [55]. Environmental cues and endogenous signals affect and determine the expression of genes associated with the initiation and development of floral organs [56]. One of the well-known classes of plant hormones, the brassinosteroids (BRs), also regulates multiple aspects of plant development, and recent evidence suggests that BRs stimulate flowering by reducing transcript levels of a potent floral repressor [57,58]. XM_030637534.1 encodes putative transcription factor BIM1 and is involved in brassinosteroid signaling [59]. Identifying these substrates and learning their regulation mechanisms by HDZ IV transcription factors would be a promising endeavor.

### 2.8. Subcellular Localization of Specific HDZ IV Gene PROTODERMAL FACTOR 2 (XP_030492336.1) in Arabidopsis Protoplast

Transcription factors are mainly localized in the nucleus. To study the subcellular localization of cannabis HDZ IV genes, we selected PROTODERMAL FACTOR 2 (XP_030492336.1) as our candidate. Cannabis PROTODERMAL FACTOR 2 (XP_030492336.1)’s homolog in tomato, PDF2, is localized in the nucleus and plasma membrane [31]. We inserted cannabis PROTODERMAL FACTOR 2 (XP_030492336.1) into plasmid pK7YWG2 and transiently transformed Arabidopsis protoplasts with the resulting plasmid. Figure 9A shows that cannabis PROTODERMAL FACTOR 2 (XP_030492336.1) is localized in the nucleus. Similar studies will research other parts of cannabis HDZ IV genes.

## 3. Discussions

The study of the functions of plant-specific HDZ IV gene family members in model plants has mainly focused on their roles in the initiation and developmental regulation of epidermal structures, such as cuticle, stomata, root hairs, and especially trichomes, a well-known “bio-factory” generating multiple secondary metabolites in cannabis [60]. These epidermal structures participate in plant development and responses to biotic and abiotic stresses. Stomata play central roles in gas exchange and water evaporation [61]. The cuticle and trichomes affect plants’ responses to biotic stresses and ultraviolet (UV) exposure. The HDZ IV gene family is considered one of the most important regulation mechanisms for sessile plants [10]. In this study, we carried out a bioinformatics survey relative to this gene family in the important crop cannabis. For past cannabis studies, most attention has been focused on the identification of novel secondary metabolites and the functional analysis of secondary metabolites (such as CBD or THC) in human health [62]. There are several methods to improve the content of secondary metabolites during the maturation period [63]. These in vitro strategies include the usage of different light conditions and plant growth effectors. Plant hormones could help cannabis flowers generate more CBD at the late flower period (unpublished data). However, drawbacks may occur when these strategies are used. It is better to induce plants to generate more secondary metabolites. Molecular biology and associated experimental strategies could help to more deeply understand the mechanisms of trichome initiation and secondary metabolites synthesis and transportation.

Our genome-wide analysis identified nine HDZ IV genes in cannabis (Table 1). The number of HDZ IV genes in cannabis is lower than that reported in other plants. Conserved domains analysis revealed that cannabis HDZ IV proteins contain all the conserved motifs, including homeodomain and START domains, which are characteristics of HDZ transcription factors. Homeodomain and the START domain may help cannabis HDZ IV proteins to functionally regulate their target genes’ expression and affect the specific phenotype. To build on these findings and to study potential benefits, future work may involve specific cannabis HDZ IV protein substrates using ChIP-seq and other strategies. Phylogenetic and gene structure analysis could enable us to decipher their function-based on homology relative to genes in better-characterized plants. Cannabis is diploid (2*n* = 20), and the genome size is about 830 Mb [32], which is six-times larger than the model plant *Arabidopsis* and two-times larger than the model plant rice. Interestingly, the number of HDZ IV genes in cannabis is smaller than in these two model plants. One possibility is that cannabis HDZ IV genes are multifunctional in regulating epidermal structures development. Cannabis HDZ IV genes share similar exon–intron architecture, with the exception of *CsROC8-like* (XP_030509195.1), which does not have 5′ and 3′ UTR. Alternative splicing is a normal phenomenon in eukaryotes and greatly increases the diversity of proteins that can be encoded by the genome. More results based on high-throughput RNA sequencing need to be analyzed before we may conclude that cannabis HDZ IV genes have different transcripts for generating multiple proteins.

Plants experience complex molecular regulation during juvenile and mature stages [64]. Transcription factors are pivotal in these stages, activating or inhibiting the expression of specific genes [65]. On the other hand, transcription factors themselves are also regulated by other regulatory proteins and/or plant hormones. Predicted promoter sequence analysis could help us to understand possible regulation mechanisms. Figure 5 shows that the promoter region of all cannabis HDZ IV genes contain light-related regulatory domains. Light regulates plant growth and development. A myriad studies demonstrated that light-related regulation illuminates every corner of plant science [66]. Trichomes are the important tissues that protect plants from UV light. Meanwhile, the daily and seasonal variation in light exposure could trigger the transition from the vegetative stage to the reproductive stage [67,68]. We can postulate that the light-associated regulators could bind specific cannabis HDZ IV gene promoter(s) and activate gene expression, which could initiate trichome development when the cannabis plants experience a short-day period [69]. Cannabis HDZ IV gene promoters also have hormone-associated binding domains for salicylic acid, methyl jasmonate, and gibberellin, which are involved in the induction of trichomes [70,71]. Different HDZ IV genes have different expression patterns in root, stem, leaf, and flower tissues. Four genes are primarily expressed in flowers and may play roles in cannabis flower trichomes’ initiation and development. Based on the expression results, we found that cannabis *Cs**HDG5* (XP_030501222.1) is mainly expressed in flower tissue and has its highest expression level in week 3 and week 4 during the flower’s mature period. Interestingly, *CsHDG5* could be induced under specific stimuli in trichomes. Some putative targets regulated by HDZ IV transcription factors are involved in pathogen signaling. *CsHDG5*, one of the HDZ IV genes, may be important to both trichome development and pathogen response.

## 4. Materials and Methods

### 4.1. Computational Identification and Analysis of Cannabis HDZ IV Genes

The proteins corresponding to the HDZ IV genes of *Oryza sativa* and *Arabidopsis* were used as queries in BLASTp searches in the cannabis genome database (assembly number: GCA_900626175.2) [37]. The cannabis proteins resulting from each blast search (E-value < 10^−5^) were pooled, and redundant sequences were removed. Protein sequences containing MEKHLA domain were also removed. Finally, the analysis of intron and exon composition was carried out by using TBtools [44]. The prediction of molecular weight and other physicochemical properties was performed by using online software ExPASy Proteomics (http://web.expasy.org/compute_pi accessed on 13 April 2021). The identification of conserved domains among cannabis HDZ IV proteins was performed using the MEME tool (http://meme-suite.org/tools/meme accessed on 7 May 2015) [41]. The identified motifs, as represented by logos, were manually inspected for the presence of elements representing conserved motifs of HDZ IV proteins.

### 4.2. Alignment, Phylogenetic Analysis and Chromosomal Localization

The multiple sequence alignment of HDZ IV sequences was performed by using MEGA11 software [72]. The phylogenetic tree was constructed by using the Maximum Likelihood method. The chromosomal mapping of individual HDZIV genes was carried out by TBtools.

### 4.3. Prediction of Cis-Regulatory Elements

In order to identify cis-regulatory domains in each cannabis HDZ IV gene’s promoter sequence, a 2 kb region upstream to the translation start codon was extracted using TBtools. These 2 kb sequences were uploaded into the Plant Cis-Acting Regulatory Element (PlantCARE) website (http://bioinformatics.psb.ugent.be/webtools/plantcare/html/ accessed on 1 January 2002) [73]. The key regulatory elements were selected manually and constructed by TBtools.

### 4.4. Plant Material and Various Treatments

“Stormy Daniels” (SD) and “Space Candy” (SC) varieties were used for gene expression analysis in the present study. The flower tissues from week 1 to week 7 were harvested from SD cannabis plants, and different tissues (root, stem, leaf, and flower) were collected from SC variety and stored at −80 °C following freezing in liquid nitrogen. In each case of flowering stages and different tissues, samples were collected in triplicate.

### 4.5. Total RNA Isolation, cDNA Synthesis, and Gene Expression Analysis

Total RNA was isolated from SD flower tissues and SC different tissues according to the previously described protocol [74]. Each RNA sample was treated with DNase to eliminate DNA contamination. The integrity and size distribution of total RNA was analyzed by agarose gel electrophoresis. A Nanodrop^®^ (Waltham, MA, USA) instrument was used to measure RNA yield (ng/µL) and purity (260:280 wavelength ratios). DNA-free RNA (2 µg) was used for synthesis of first strand cDNA by using a Bio-Rad iScript™ (Hercules, CA, USA) cDNA synthesis kit as per manufacturer’s recommendations. Quantitative real-time PCR was performed with volumes of 20 µL per well with Bio-Rad™ SYBR green Supermix (Hercules, CA, USA). The amount of cDNA was normalized by using an amplification of housekeeping cannabis actin as an internal control. The data from real-time PCR amplification were estimated in terms of comparative fold expression following the 2^(2−ΔΔct)^ method [75]. The list of different primers used in the study is provided in Appendix A.

### 4.6. Subcellular Localization of PROTODERMAL FACTOR 2 (XP_030492336.1)

*PROTODERMAL FACTOR 2* (XP_030492336.1) was fused to EYFP using a 35S promoter-containing plasmid (pK7YWG2) with LR Clonase II (Invitrogen, Waltham, MA, USA), and the resulting construct was transiently transformed into *Arabidopsis* protoplast [76]. Protoplasts were viewed using a Nikon A1R confocal microscope through a 20× Plan Apo lens. Both channels were excited at 514 nm. Emissions were collected with an EYFP filter (514 nm). Z-stacks were collected at a step size of 27 microns. Composite channel/stack images/scale bars were produced in ImageJ [77].

## 5. Conclusions

This work used bioinformatics strategies to identify the HDZ IV gene family in *Cannabis sativa* for the first time. This gene family is central in regulating epidermal structures such as trichomes, which generate multiple secondary metabolites. Understanding the molecular mechanisms involved in these regulation networks will help researchers to construct genetically modified plants that generate more trichomes in every tissue. Indeed, transcription factors could be an ideal regulatory tool to accomplish this task. Thus, our results represent a useful foundation for developing an approach in cannabis research.

## Figures and Tables

**Figure 1 plants-11-01307-f001:**
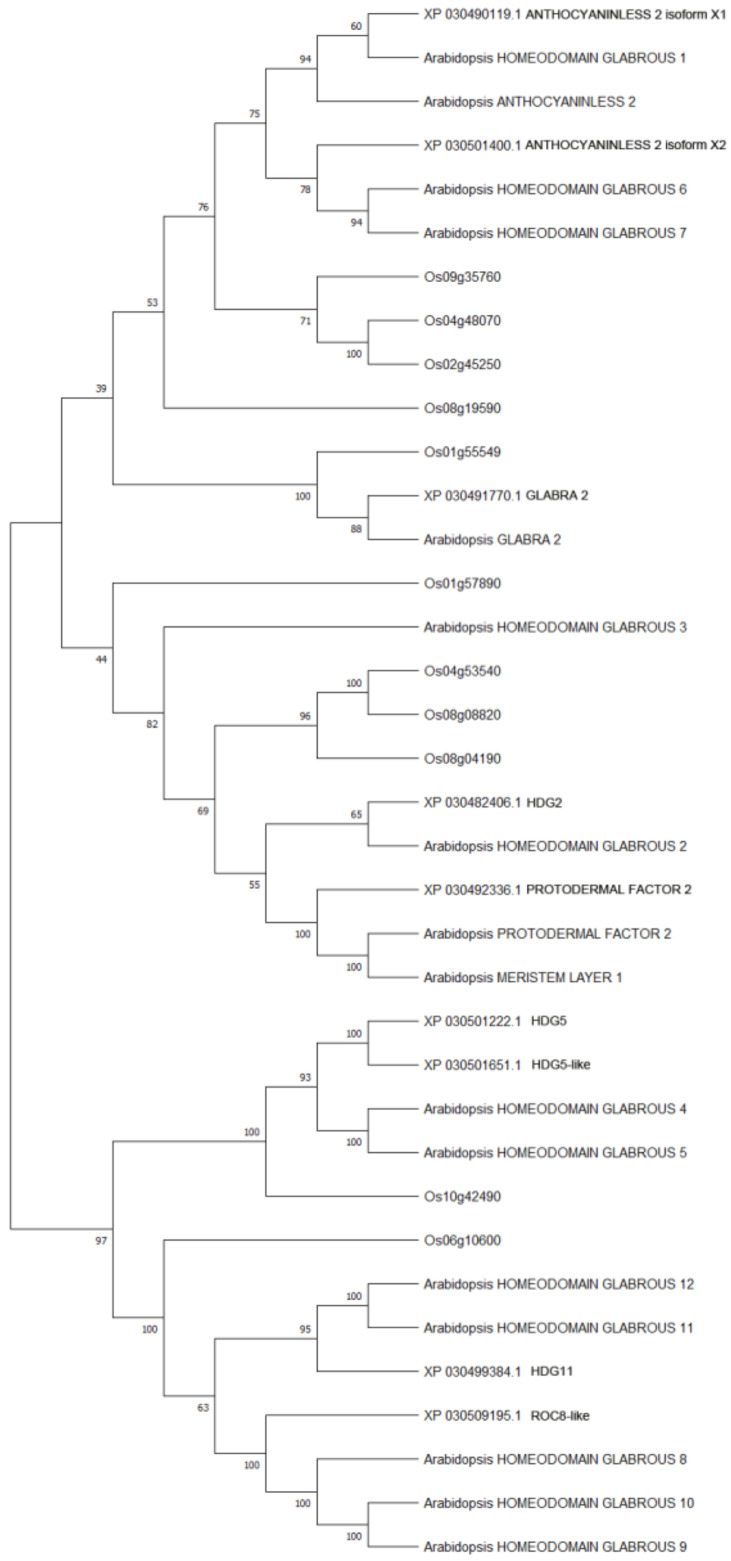
Molecular phylogenetic analysis of cannabis HDZ IV proteins and model plants HDZ IV proteins by Maximum Likelihood method. The evolutionary history was inferred by using the Maximum Likelihood method based on the JTT matrix-based model [39]. The bootstrap consensus tree inferred from 500 replicates [40] is taken to represent the evolutionary history of the taxa analyzed [40]. Branches corresponding to partitions reproduced in less than 50% bootstrap replicates are collapsed. The percentage of replicate trees in which the associated taxa clustered together in the bootstrap test (500 replicates) is shown next to the branches [40].

**Figure 2 plants-11-01307-f002:**
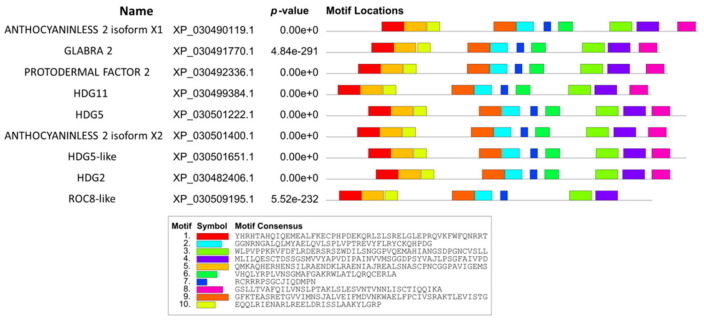
Schematic representation of conserved motifs in cannabis HDZ IV proteins.

**Figure 3 plants-11-01307-f003:**
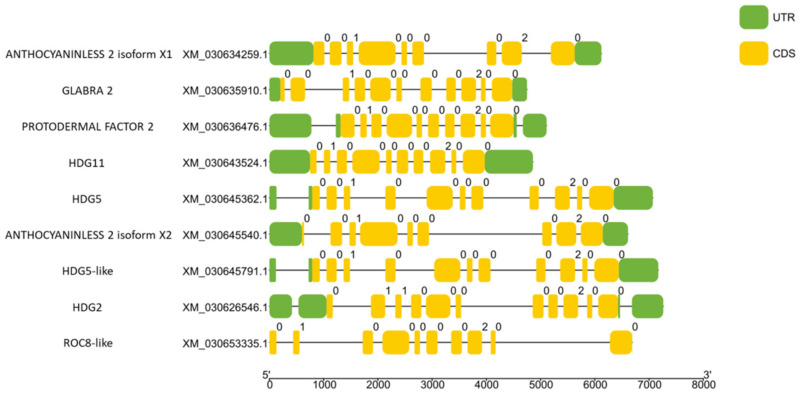
Schematic representation of UTR and intron–exon composition of cannabis HDZ IV genes. Exon end phase 0 indicates no interruption; exon end phase 1 indicates last codon’s last base is in the next exon; exon end phase 2 indicates that the last codon’s last two bases are in the next exon.

**Figure 4 plants-11-01307-f004:**
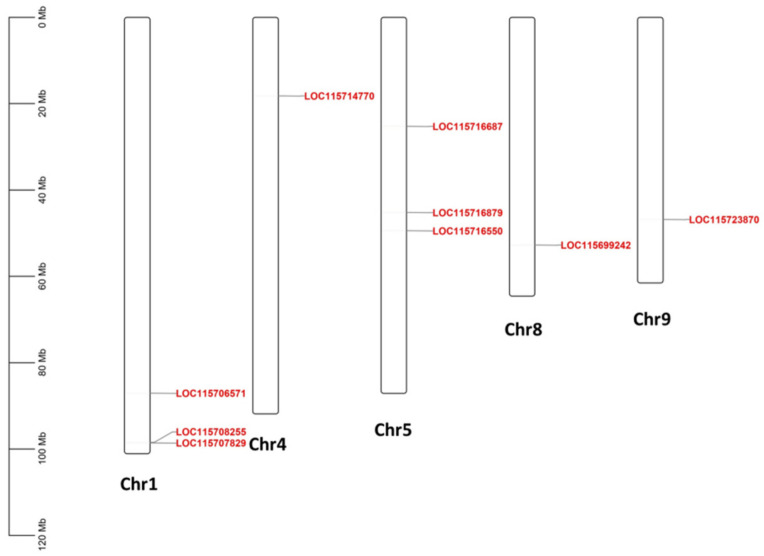
Chromosomal localization of cannabis HDZ IV genes.

**Figure 5 plants-11-01307-f005:**
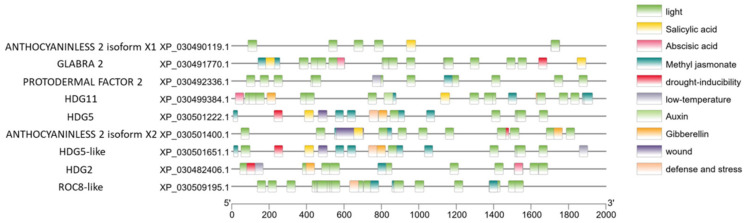
Cis-regulatory elements present in the promoters of cannabis HDZ IV genes.

**Figure 6 plants-11-01307-f006:**
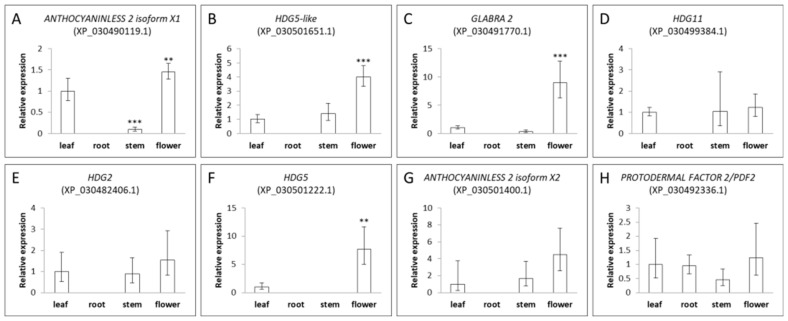
Expression pattern of eight cannabis HDZ IV genes in different tissues of Space Candy (SC). (**A**) the expression pattern of *ANTHOCYANINLESS 2 isoform X1* (XP_030490119.1) in different tissues; (**B**) the expression pattern of *HDG5-like* (XP_030501651.1) in different tissues; (**C**) the expression pattern of *GLABRA2* (XP_030491770.1) in different tissues; (**D**) the expression pattern of *HDG11* (XP_030499384.1) in different tissues; (**E**) the expression pattern of *HDG2* (XP_030482406.1) in different tissues; (**F**) the expression pattern of *HDG5* (XP_030501222.1) in different tissues; (**G**) the expression pattern of *ANTHOCYANINLESS 2 isoform X2* (XP_030501400.1) in different tissues; (**H**) the expression pattern of *PROTODERMAL FACTOR 2/PDF2* (XP_030492336.1) in different tissues. The ∆∆CT method was used for calculations. *X*-axis refers to different tissues. Data are presented as means ± SE (*n* = 3). Means separation between expression in various tissues compared to the level in leaf was evaluated using Student’s *t*-test; ** indicates *p* < 0.01, and *** indicates *p* < 0.001.

**Figure 7 plants-11-01307-f007:**
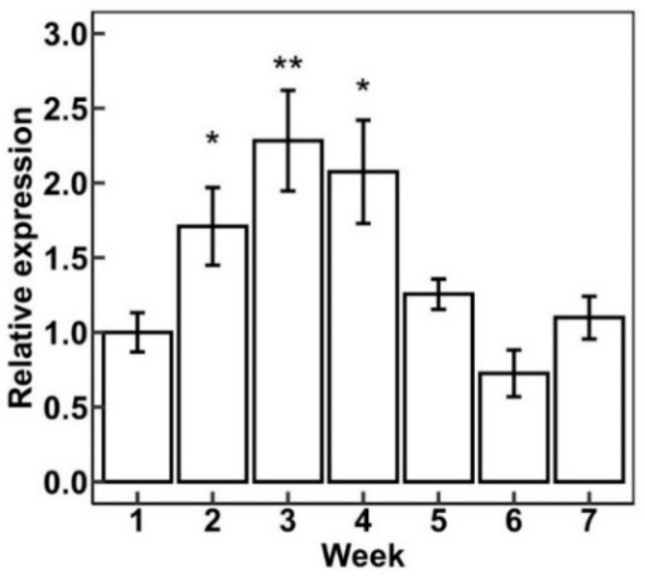
Expression pattern of Cannabis HDG5 (XP_030501222.1) during Stormy Daniels (SD) flower maturity stages. The ∆∆CT method was used for calculations. *X*-axis refers to weeks after the change from 18 h: 6 h light to 12 h: 12 h light. Data are presented as means ± SE (*n* = 4). Means separation between expression at various times during flower development as compared to expression at week one was evaluated using Student’s *t*-test; * indicates *p* < 0.05, and ** indicates *p* < 0.01.

**Figure 8 plants-11-01307-f008:**
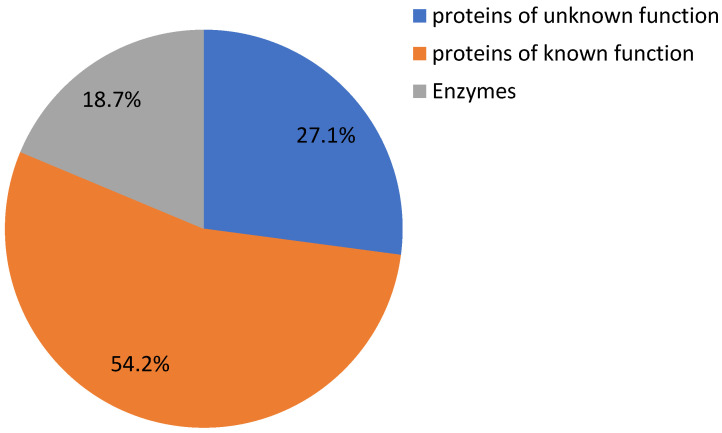
Classification of putative target genes regulated by HDZ IV transcription factors.

**Figure 9 plants-11-01307-f009:**
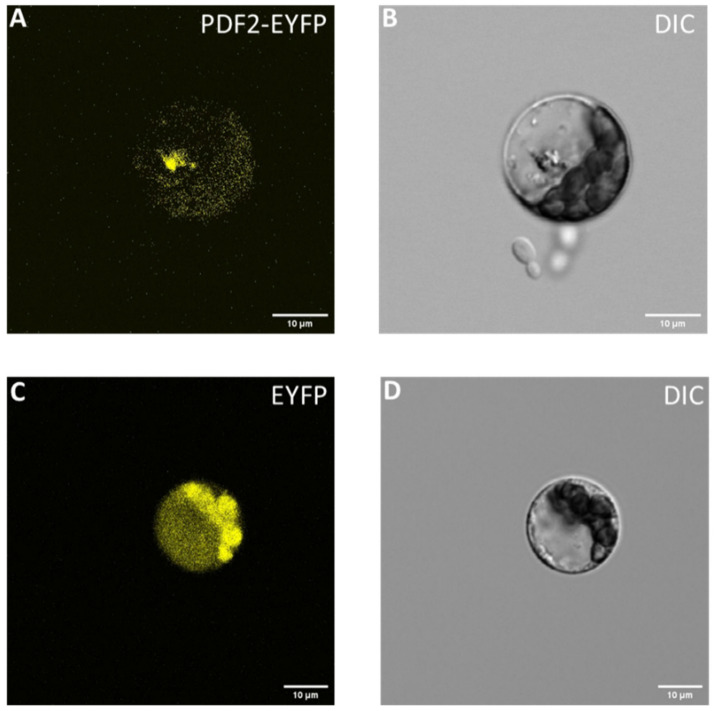
Subcellular localization of cannabis PROTODERMAL FACTOR 2 (XP_030492336.1) gene. (**A**) PDF2 was localized in the *Arabidopsis* protoplast nucleus. (**B**) DIC of *Arabidopsis* protoplast. (**C**) *Arabidopsis* protoplasts transiently transformed with pK7YWG2. (**D**) DIC of *Arabidopsis* protoplast.

**Table 1 plants-11-01307-t001:** Structural features of *CsHDZIV* genes in cannabis.

Gene	Gene ID	mRNA Locus	Length (aa)	MW (Da)	pI	Grand Average of Hydropathicity (GRAVY)	Instability Index	Aliphatic Index	Subcellular Location
LOC115706571 (ANTHOCYANINLESS 2 isoform X1)	XP_030490119.1	XM_030634259.1	841	91,400.22	5.69	−0.345	52.84	77.35	nucl: 13, pero: 1
LOC115707829 (GLABRA 2)	XP_030491770.1	XM_030635910.1	755	83,575.46	5.64	−0.457	48.05	74.78	nucl: 12, extr: 2
LOC115708255 (PROTODERMAL FACTOR 2)	XP_030492336.1	XM_030636476.1	772	84,561.23	5.81	−0.364	40.27	81.09	nucl: 14
LOC115714770 (HDG11)	XP_030499384.1	XM_030643524.1	734	79,965.59	6.33	−0.328	54.15	77.18	nucl: 14
LOC115716550 (HDG5)	XP_030501222.1	XM_030645362.1	814	90,647.37	5.62	−0.458	52.88	74.75	nucl: 12, extr: 2
LOC115716687 (ANTHOCYANINLESS 2 isoform X2)	XP_030501400.1	XM_030645540.1	776	84,873.60	5.68	−0.282	47.12	80.55	nucl: 13, pero: 1
LOC115716879 (HDG5-like)	XP_030501651.1	XM_030645791.1	814	90,670.40	5.66	−0.458	52.17	74.75	nucl: 12, extr: 2
LOC115699242 (HDG2)	XP_030482406.1	XM_030626546.1	784	85,192.34	5.75	−0.296	41.12	78.72	nucl: 12, extr: 2
LOC115723870 (ROC8-like)	XP_030509195.1	XM_030653335.1	737	81,729.85	6.41	−0.329	49.62	87.14	nucl: 11, cyto: 2, vacu: 1

## Data Availability

Not applicable.

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
