# Peer review of "Genome-Wide Identification and Expression Analysis of Homeodomain Leucine Zipper Subfamily IV (HD-ZIP IV) Gene Family in Cannabis sativa L."

_plants, 2022, doi:10.3390/plants11101307_

Round 1

Reviewer 1 Report

Cannabis generates produces known and unknown secondary metabolites. This study focused on increasing the number of glandular trichomes in leaves to promote the content of these secondary metabolites. They identified the HDZ IV family in cannabis and performed a systematic analysis of this family. Notably, the gene candidates for genetic improvement were raised in this study.
1. The candidate genes of this study were raised by the expression analysis, while the expression profile of other genes seems to be absent, except CaHDG5. Please add the expression profile of other members by qRT-PCR, which provides more clues to support your conclusions.
2. The cannabis HDZ IV genes should be named according to their chromosome location information as the previous identification study.
3. All genes should be italic like the HDZ IV gene in Line19, as well as the Arabidopsis in Line 48.
4. The MEKHLA domain number of Pfam and Smart should be listed in Line 295.
5. In Line 305, authors should opt for the latest version of MEGA like MEGA 11.
6. In Line 332, it should be the 2-ΔΔct was adopted.
7. The content in Line 224-227 should be removed.

Author Response

Comments from Reviewer 1

Cannabis generates produces known and unknown secondary metabolites. This study focused on increasing the number of glandular trichomes in leaves to promote the content of these secondary metabolites. They identified the HDZ IV family in cannabis and performed a systematic analysis of this family. Notably, the gene candidates for genetic improvement were raised in this study.

  • Comment 1: The candidate genes of this study were raised by the expression analysis, while the expression profile of other genes seems to be absent, except CsHDG5. Please add the expression profile of other members by qRT-PCR, which provides more clues to support your conclusions.

     Response: Thank you for pointing this out. We agree with this comment. Gene expression analysis could give us a ‘lead’ into better understanding the function of members of the gene family. Therefore, we have performed qRT-PCR for all genes in different tissues (leaf, root, stem and flower). The results are shown in Figure 6 of the revised ms. Notably, in addition to the expression of CsHDG5, we note from the new experimental results (Fig. 6), that the HDZ IV genes ‘HDG5-like’ and ‘GLABRA 2’ are also expressed in cannabis flowers to a greater extent than other tissues/organs. We discuss these new experimental results starting on line 192 (the last paragraph on page 6)/

  • Comment 2: The cannabis HDZ IV genes should be named according to their chromosome location information as the previous identification study.

     Response: Agree. We have revised all HDZ IV genes mentioned in the manuscript.

  • Comment 3: All genes should be italic like the HDZ IV gene in Line19, as well as the Arabidopsis in Line 48. The genes were italic in Line19, as well as the Arabidopsis in Line 48.

     Response: Thanks for pointing this out. We have revised all genes to appear in italics font in the revised manuscript.

  • Comment 4: The MEKHLA domain number of Pfam and Smart should be listed in Line 295.

     Response: We agree with this. HDZ III gene family has the MEKHLA domain. So we removed three candidate genes with MEKHLA domain. We have incorporated your suggestion throughout the manuscript. The analysis of the domain of the HDZ IV family members is shown in Supplemental Figure S1 in the revised manuscript.

  • Comment 5: In Line 305, authors should opt for the latest version of MEGA like MEGA 11.

     Response: Thanks for pointing this out. We used MEGA11 to do the molecular phylogenetic analysis and revised Figure 1 correspondingly.

  • Comment 6: In Line 332, it should be the 2-ΔΔct was adopted.

     Response: Thanks for pointing this out. We revised 2-ΔΔct within the revised manuscript.

  • Comment 7: The content in Line 224-227 should be removed.

     Response: Agree. We removed the sentences within the revised manuscript.

Reviewer 2 Report

The manuscript entitled “Genome-wide identification and expression analysis of Homeodomain leucine Zipper subfamily IV (HD-ZIP IV) Gene Family in Cannabis sativa L ” by Gang Ma, et al describes nine homeodomain zipper family transcription factors IV (HDZ IV) genes through the computational analysis and experiments. 

First, the authors use bioinformatics to find these genes putatively encoding proteins with the conserved domains of HDZ IV transcription factors. The phylogenetic analysis implies that those genes might follow distinct evolutionary paths after divergence from a common ancestor. All the identified HDZ IV gene promoter sequences have multiple regulation motifs, such as light- and hormone-responsive elements. Then they experimentally study the expression pattern of CsHDG5 that were correlated with flowering maturity. Those results provide a potential solution to increase the number of cannabis trichomes and improve secondary metabolites in the future. 

Overall, hypotheses are appropriate. The major concern about this study is the lack of experimental evidence to support that HDZ IV genes encode transcription factors. For example, authors should construct HDZ IV mutants and then discuss the differences of the expression patterns between wild type and mutants. And how they can be interpreted from the perspective of the hypotheses that HDZ IV genes are correlated with flowering maturity. 

I have highlighted some minor concerns as below:

1 Line 64 “Artemisia annua” should be italic; 

Similarly, Line 65 “Solanum lycopersicum”;

Line 119, “in vitro”, “in vivo”

2 Although the authors describe the numbers of putative targets of HDZ IV transcription factors in Line199, it is necessary to label the percentage or the numbers for each type of putative targets in figure 7. Meanwhile, it is necessary to list 59 putative targets in the supplementary materials. 

Author Response

Comments from Reviewer 2

The manuscript entitled “Genome-wide identification and expression analysis of Homeodomain leucine Zipper subfamily IV (HD-ZIP IV) Gene Family in Cannabis sativa L ” by Gang Ma, et al describes nine homeodomain zipper family transcription factors IV (HDZ IV) genes through the computational analysis and experiments. 

First, the authors use bioinformatics to find these genes putatively encoding proteins with the conserved domains of HDZ IV transcription factors. The phylogenetic analysis implies that those genes might follow distinct evolutionary paths after divergence from a common ancestor. All the identified HDZ IV gene promoter sequences have multiple regulation motifs, such as light- and hormone-responsive elements. Then they experimentally study the expression pattern of CsHDG5 that were correlated with flowering maturity. Those results provide a potential solution to increase the number of cannabis trichomes and improve secondary metabolites in the future. 

  • Comment 1: Overall, hypotheses are appropriate. The major concern about this study is the lack of experimental evidence to support that HDZ IV genes encode transcription factors. For example, authors should construct HDZ IV mutants and then discuss the differences of the expression patterns between wild type and mutants. And how they can be interpreted from the perspective of the hypotheses that HDZ IV genes are correlated with flowering maturity.

     Response: Thank you for pointing this out. It is useful to study gene function using overexpression and knock out strategies. However, cannabis is recalcitrant to stable transformation. We used an alternative strategy (subcellular localization) to support our hypothesis. Figure 9 shows new experimental results undertaken to address the reviewer’s point. We cloned the one of the cannabis HDZ IV gene family members, PROTODERMAL FACTOR 2 (XP_030492336.1) (PDF2) and expressed the translation product in Arabidopsis protoplasts. As shown in Figure 9, a fusion construct of PDF2 generated with an YFP tag was localized in the nucleus. The control, YFP expressed alone, was localized in numerous tissues but not the nucleus. This new evidence could partly verify our hypothesis. We did qPCR for all HDZ IV gene family members and found that several members of this gene family are highly expressed in flowers. Our purpose to study this gene family is to increase identify molecular mechanisms that may impact trichome formation in the flower tissue of cannabis. So we chose one of the genes (HDG5) highly expressed in cannabis female flowers as our candidate gene. Figure 7 in the revised manuscript shows the expression pattern of HDG5 during female flower development; HDG5 expression increases during flower development.

  • Comment 2: 1 Line 64 “Artemisia annua” should be italic;

Similarly, Line 65 “Solanum lycopersicum”;

Line 119, “in vitro”, “in vivo”

     Response: Thanks for pointing this out. We revised Artemisia annua, Solanum lycopersicum and “in vivo” in lines 67, 68 and 124, respectively, of the revised manuscript.

  • Comment 3: Although the authors describe the numbers of putative targets of HDZ IV transcription factors in Line199, it is necessary to label the percentage or the numbers for each type of putative targets in figure 7. Meanwhile, it is necessary to list 59 putative targets in the supplementary materials

     Response: Agree. We revised the figure 8 and added Supplementary Table 2 listing 59 putative targets in the revised manuscript.

Round 2

Reviewer 1 Report

The current revised manuscript has many improvements in the elucidation of cannabis HD-ZIP families. The manuscript can be accepted in the current version.